# AWAC: Accelerating Online Reinforcement Learning with Offline Datasets

## Abstract

Reinforcement learning provides an appealing formalism for learning control policies from experience. However, the classic active formulation of reinforcement learning necessitates a lengthy active exploration process for each behavior, making it difficult to apply in real-world settings. If we can instead allow reinforcement learning to effectively use previously collected data to aid the online learning process, where the data could be expert demonstrations or more generally any prior experience, we could make reinforcement learning a substantially more practical tool. While a number of recent methods have sought to learn offline from previously collected data, it remains exceptionally difficult to train a policy with offline data and improve it further with online reinforcement learning. In this paper we systematically analyze why this problem is so challenging, and propose an algorithm that combines sample-efficient dynamic programming with maximum likelihood policy updates, providing a simple and effective framework that is able to leverage large amounts of offline data and then quickly perform online fine-tuning of reinforcement learning policies. We show that our method enables rapid learning of skills with a combination of prior demonstration data and online experience across a suite of difficult dexterous manipulation and benchmark tasks.

## 1 Introduction

Learning models that generalize effectively to complex open-world settings, from image recognition (Krizhevsky et al., 2012) to natural language processing (Devlin et al., 2019), relies on large, high-capacity models and large, diverse, and representative datasets. Leveraging this recipe for reinforcement learning (RL) has the potential to yield real-world generalization for control applications such as robotics. However, while deep RL algorithms enable the use of large models, the use of large datasets for real-world RL has proven challenging. Most RL algorithms collect new data online every time a new policy is learned, which limits the size and diversity of the datasets for RL. In the same way that powerful models in computer vision and NLP are often pre-trained on large, general-purpose datasets and then fine-tuned on task-specific data, RL policies that generalize effectively to open-world settings will need to be able to incorporate large amounts of prior data effectively into the learning process, while still collecting additional data online for the task at hand.

For data-driven reinforcement learning, offline datasets consist of trajectories of states, actions and associated rewards. This data can potentially come from demonstrations for the desired task (Schaal, 1997; Atkeson & Schaal, 1997), suboptimal policies (Gao et al., 2018), demonstrations for related tasks (Zhou et al., 2019), or even just random exploration in the environment. Depending on the quality of the data that is provided, useful knowledge can be extracted about the dynamics of the world, about the task being solved, or both. Effective data-driven methods for deep reinforcement learning should be able to use this data to pre-train offline while improving with online fine-tuning.

Since this prior data can come from a variety of sources, we would like to design an algorithm that does not utilize different types of data in any privileged way. For example, prior methods that incorporate demonstrations into RL directly aim to mimic these demonstrations (Nair et al., 2018), which is desirable when the demonstrations are known to be optimal, but imposes strict requirements on the type of offline data, and can cause undesirable bias when the prior data is not optimal. While prior methods for fully offline RL provide a mechanism for utilizing offline data (Fujimoto et al., 2019; Kumar et al., 2019), as we will show in our experiments, such methods generally are not effective for fine-tuning with online data as they are often too conservative. In effect, prior methods

require us to choose: Do we assume prior data is optimal or not? Do we use only offline data, or only online data? To make it feasible to learn policies for open-world settings, we need algorithms that learn successfully in any of these cases.

In this work, we study how to build RL algorithms that are effective for pre-training from off-policy datasets, but also well suited to continuous improvement with online data collection. We systematically analyze the challenges with using standard off-policy RL algorithms (Haarnoja et al., 2018; Kumar et al., 2019; Abdolmaleki et al., 2018) for this problem, and introduce a simple actor critic algorithm that elegantly bridges data-driven pre-training from offline data and improvement with online data collection. Our method, which uses dynamic programming to train a critic but a supervised learning style update to train a constrained actor, combines the best of supervised learning and actor-critic algorithms. Dynamic programming can leverage off-policy data and enable sample-efficient learning. The simple supervised actor update implicitly enforces a constraint that mitigates the effects of distribution shift when learning from offline data (Fujimoto et al., 2019; Kumar et al., 2019), while avoiding overly conservative updates.

We evaluate our algorithm on a wide variety of robotic control and benchmark tasks across three simulated domains: dexterous manipulation, tabletop manipulation, and MuJoCo control tasks. Our algorithm, Advantage Weighted Actor Critic (AWAC), is able to quickly learn successful policies on difficult tasks with high action dimension and binary sparse rewards, significantly better than prior methods for off-policy and offline reinforcement learning. Moreover, AWAC can utilize different types of prior data without any algorithmic changes: demonstrations, suboptimal data, or random exploration data. The contribution of this work is not just another RL algorithm, but a systematic study of what makes offline pre-training with online fine-tuning unique compared to the standard RL paradigm, which then directly motivates a simple algorithm, AWAC, to address these challenges.

## 2 Preliminaries

We consider the standard reinforcement learning notation, with states $\mathbf{s}$, actions $\mathbf{a}$, policy $\pi(\mathbf{a}|\mathbf{s})$, rewards $r(\mathbf{s}, \mathbf{a})$, and dynamics $p(\mathbf{s}'|\mathbf{s}, \mathbf{a})$. The discounted return is defined as $R_t = \sum_{i=t}^{T} \gamma^i r(\mathbf{s}_i, \mathbf{a}_i)$, for a discount factor $\gamma$ and horizon $T$ which may be infinite. The objective of an RL agent is to maximize the expected discounted return $J(\pi) = \mathbb{E}_{p_\pi(\tau)}[R_0]$ under the distribution induced by the policy. The optimal policy can be learned directly by policy gradient, estimating $\nabla J(\pi)$ (Williams, 1992), but this is often ineffective due to high variance of the estimator. Many algorithms attempt to reduce this variance by making use of the value function $V^\pi(\mathbf{s}) = \mathbb{E}_{p_\pi(\tau)}[R_t|\mathbf{s}]$, action-value function $Q^\pi(\mathbf{s}, \mathbf{a}) = \mathbb{E}_{p_\pi(\tau)}[R_t|\mathbf{s}, \mathbf{a}]$, or advantage $A^\pi(\mathbf{s}, \mathbf{a}) = Q^\pi(\mathbf{s}, \mathbf{a}) - V^\pi(\mathbf{s})$. The action-value function for a policy can be written recursively via the Bellman equation:

$$Q^\pi(\mathbf{s}, \mathbf{a}) = r(\mathbf{s}, \mathbf{a}) + \gamma \mathbb{E}_{p(\mathbf{s}'|\mathbf{s}, \mathbf{a})}[V^\pi(\mathbf{s}')] = r(\mathbf{s}, \mathbf{a}) + \gamma \mathbb{E}_{p(\mathbf{s}'|\mathbf{s}, \mathbf{a})}[\mathbb{E}_{\pi(\mathbf{a}'|\mathbf{s}')}[Q^\pi(\mathbf{s}', \mathbf{a}')]]. \quad (1)$$

Instead of estimating policy gradients directly, actor-critic algorithms maximize returns by alternating between two phases (Konda & Tsitsiklis, 2000): policy evaluation and policy improvement. During the policy evaluation phase, the critic $Q^\pi(\mathbf{s}, \mathbf{a})$ is estimated for the current policy $\pi$. This can be accomplished by repeatedly applying the Bellman operator $\mathcal{B}$, corresponding to the right-hand side of Equation 1, as defined below:

$$\mathcal{B}^\pi Q(\mathbf{s}, \mathbf{a}) = r(\mathbf{s}, \mathbf{a}) + \gamma \mathbb{E}_{p(\mathbf{s}'|\mathbf{s}, \mathbf{a})}[\mathbb{E}_{\pi(\mathbf{a}'|\mathbf{s}')}[Q^\pi(\mathbf{s}', \mathbf{a}')]]. \quad (2)$$

By iterating according to $Q^{k+1} = \mathcal{B}^\pi Q^k$, $Q^k$ converges to $Q^\pi$ (Sutton & Barto, 1998). With function approximation, we cannot apply the Bellman operator exactly, and instead minimize the Bellman error with respect to Q-function parameters $\phi_k$:

$$\phi_k = \arg\min_\phi \mathbb{E}_\mathcal{D}[(Q_\phi(\mathbf{s}, \mathbf{a}) - y)^2], y = r(\mathbf{s}, \mathbf{a}) + \gamma \mathbb{E}_{\mathbf{s}', \mathbf{a}'}[Q_{\phi_{k-1}}(\mathbf{s}', \mathbf{a}')]. \quad (3)$$

During policy improvement, the actor $\pi$ is typically updated based on the current estimate of $Q^\pi$. A commonly used technique (Lillicrap et al., 2016; Fujimoto et al., 2018; Haarnoja et al., 2018) is to update the actor $\pi_{\theta_k}(\mathbf{a}|\mathbf{s})$ via likelihood ratio or pathwise derivatives to optimize the following objective, such that the expected value of the Q-function $Q^\pi$ is maximized:

$$\theta_k = \arg\max_\theta \mathbb{E}_{\mathbf{s} \sim \mathcal{D}}[\mathbb{E}_{\pi_\theta(\mathbf{a}|\mathbf{s})}[Q_{\phi_k}(\mathbf{s}, \mathbf{a})]] \quad (4)$$

Actor-critic algorithms are widely used in deep RL (Mnih et al., 2016; Lillicrap et al., 2016; Haarnoja et al., 2018; Fujimoto et al., 2018). With a Q-function estimator, they can in principle utilize off-policy data when used with a replay buffer for storing prior transition tuples, which we will denote $\beta$, to sample previous transitions, although we show that this by itself is insufficient for our problem setting.

Figure 1: We study learning policies by offline learning on a prior dataset $\mathcal{D}$ and then fine-tuning with online interaction. The prior data could be obtained via prior runs of RL, expert demonstrations, or any other source of transitions. Our method, advantage weighted actor critic (AWAC) is able to learn effectively from offline data and fine-tune in order to reach expert-level performance after collecting a limited amount of interaction data. Videos and data are available at *sites.google.com/view/awac-anonymous*

## 3 CHALLENGES IN OFFLINE RL WITH ONLINE FINE-TUNING

In this section, we study the unique challenges that exist when pre-training using offline data, followed by fine-tuning with online data collection. We first describe the problem, and then analyze what makes this problem difficult for prior methods.

**Problem definition.** A static dataset of transitions, $\mathcal{D} = \{(\mathbf{s}, \mathbf{a}, \mathbf{s}', r)_j\}$, is provided to the algorithm at the beginning of training. This dataset can be sampled from an arbitrary policy or mixture of policies, and may even be collected by a human expert. This definition is general and encompasses many scenarios, such as learning from demonstrations, random data, prior RL experiments, or even from multi-task data. Given the dataset $\mathcal{D}$, our goal is to leverage $\mathcal{D}$ for pre-training and use some online interaction to learn the optimal policy $\pi^*(\mathbf{a}|\mathbf{s})$, with as few interactions with the environment as possible (depicted in Fig 1). This setting is representative of many real-world RL settings, where prior data is available and the aim is to learn new skills efficiently. We first study existing algorithms empirically in this setting on the HalfCheetah-v2 Gym environment[1]. The prior dataset consists of 15 demonstrations from an expert policy and 100 suboptimal trajectories sampled from a behavioral clone of these demonstrations. All methods for the remainder of this paper incorporate the prior dataset, unless explicitly labeled "scratch".

**3.1) Data Efficiency.** One of the simplest ways to utilize prior data such as demonstrations for RL is to pre-train a policy with imitation learning, and fine-tune with on-policy RL (Gupta et al., 2019; Rajeswaran et al., 2018). This approach has two drawbacks: (1) prior data may not be optimal; (2) on-policy fine-tuning is data inefficient as it does not reuse the prior data in the RL stage. In our setting, data efficiency is vital. To this end, we require algorithms that are able to reuse arbitrary off-policy data during online RL for data-efficient fine-tuning. We find that algorithms that use on-policy fine-tuning (Rajeswaran et al., 2018; Gupta et al., 2019), or Monte-Carlo return estimation (Peters & Schaal, 2007; Wang et al., 2018; Peng et al., 2019) are generally much less efficient than off-policy actor-critic algorithms, which iterate between improving $\pi$ and estimating $Q^\pi$ via Bellman backups. This can be seen from the results in Figure 2 plot 1, where on-policy methods like DAPG (Rajeswaran et al., 2018) and Monte-Carlo return methods like AWR (Peng et al., 2019) and MARWIL (Wang et al., 2018) are an order of magnitude slower than off-policy actor-critic methods. Actor-critic methods, shown in Figure 2 plot 2, can in principle use off-policy data. However, as we will discuss next, naïvely applying these algorithms to our problem suffers from a different set of challenges.

**3.2) Bootstrap Error in Offline Learning with Actor-Critic Methods.** When standard off-policy actor-critic methods are applied to this problem setting, they perform poorly, as shown in the second plot in Figure 2: despite having a prior dataset in the replay buffer, these algorithms do not benefit significantly from offline training. We evaluate soft actor critic (Haarnoja et al., 2018), a state-of-the-art actor-critic algorithm for continuous control. Note that "SAC-scratch," which does not receive the prior data, performs similarly to "SACfD-prior," which does have access to the prior data, indicating that the off-policy RL algorithm is not actually able to make use of the off-policy data for pre-training. Moreover, even if the SAC is policy is pre-trained by behavior cloning, labeled "SACfD-pretrain", we still observe an initial decrease in performance, and performance similar to learning from scratch.

This challenge can be attributed to off-policy bootstrapping error accumulation, as observed in several prior works (Sutton & Barto, 1998; Kumar et al., 2019; Wu et al., 2020; Levine et al., 2020;

---

[1]We use this environment for analysis because it helps understand and accentuate the differences between different algorithms. More challenging environments like the ones shown in Fig 3 are too hard to solve to analyze variants of different methods.

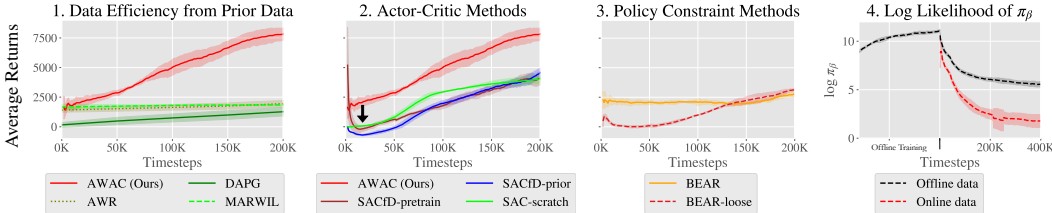

Figure 2: Analysis of prior methods on HalfCheetah-v2 using offline RL with online fine-tuning. (1) On-policy methods (DAPG, AWR, MARWIL) learn relatively slowly, even with access to prior data. We present our method, AWAC, as an example of how off-policy RL methods can learn much faster. (2) Variants of soft actor-critic (SAC) with offline training (performed before timestep 0) and fine-tuning. We see a "dip" in the initial performance, even if the policy is pretrained with behavioral cloning. (3) Offline RL method BEAR (Kumar et al., 2019) on offline training and fine-tuning, including a "loose" variant of BEAR with a weakened constraint. Standard offline RL methods fine-tune slowly, while the "loose" BEAR variant experiences a similar dip as SAC. (4) We show that the fit of the behavior models $\hat{\pi}_\beta$ used by these offline methods degrades as new data is added to the buffer during fine-tuning, potentially explaining their poor fine-tuning performance.

Fujimoto et al., 2019). In actor-critic algorithms, the target value $Q(\mathbf{s}', \mathbf{a}')$, with $\mathbf{a}' \sim \pi$, is used to update $Q(\mathbf{s}, \mathbf{a})$. When $\mathbf{a}'$ is outside of the data distribution, $Q(\mathbf{s}', \mathbf{a}')$ will be inaccurate, leading to accumulation of error on static datasets.

Offline RL algorithms (Fujimoto et al., 2019; Kumar et al., 2019; Wu et al., 2020) propose to address this issue by explicitly adding constraints on the policy improvement update (Equation 4) to avoid bootstrapping on out-of-distribution actions, leading to a policy update of this form:

$$\arg \max_{\theta} \mathbb{E}_{\mathbf{s} \sim \mathcal{D}}[\mathbb{E}_{\pi_\theta(\mathbf{a}|\mathbf{s})}[Q_{\phi_k}(\mathbf{s}, \mathbf{a})]] \text{ s.t. } D(\pi_\theta, \pi_\beta) \leq \epsilon. \tag{5}$$

Here, $\pi_\theta$ is the actor being updated, and $\pi_\beta(a|s)$ represents the (potentially unknown) distribution from which all of the data seen so far (both offline data and online data) was generated. In the case of a replay buffer, $\pi_\beta$ corresponds to a mixture distribution over all past policies. Typically, $\pi_\beta$ is not known, especially for offline data, and must be estimated from the data itself. Many offline RL algorithms (Kumar et al., 2019; Fujimoto et al., 2019; Siegel et al., 2020) explicitly fit a parametric model to samples for the distribution $\pi_\beta$ via maximum likelihood estimation, where samples from $\pi_\beta$ are obtained simply by sampling uniformly from the data seen thus far: $\hat{\pi}_\beta = \max_{\hat{\pi}_\beta} \mathbb{E}_{\mathbf{s}, \mathbf{a} \sim \pi_\beta}[\log \hat{\pi}_\beta(\mathbf{a}|\mathbf{s})]$. After estimating $\hat{\pi}_\beta$, prior methods implement the constraint given in Equation 5 in various ways, including penalties on the policy update (Kumar et al., 2019; Wu et al., 2020) or architecture choices for sampling actions for policy training (Fujimoto et al., 2019; Siegel et al., 2020). As we will see next, the requirement for accurate estimation of $\hat{\pi}_\beta$ makes these methods difficult to use with online fine-tuning.

**3.3) Excessively Conservative Online Learning.** While offline RL algorithms with constraints (Kumar et al., 2019; Fujimoto et al., 2019; Wu et al., 2020) perform well offline, they struggle to improve with fine-tuning, as shown in the third plot in Figure 2. We see that the purely offline RL performance (at "0K" in Fig. 2) is much better than the standard off-policy methods shown in Section 3.2. However, with additional iterations of online fine-tuning, the performance increases very slowly (as seen from the slope of the BEAR curve in Fig 2). What causes this phenomenon?

This can be attributed to challenges in fitting an accurate behavior model as data is collected online during fine-tuning. In the offline setting, behavior models must only be trained once via maximum likelihood, but in the online setting, the behavior model must be updated online to track incoming data. Training density models online (in the "streaming" setting) is a challenging research problem (Ramapuram et al., 2017), made more difficult by a potentially complex multi-modal behavior distribution induced by the mixture of online and offline data. To understand this, we plot the log likelihood of learned behavior models on the dataset during online and offline training for the HalfCheetah task. As we can see in the plot, the accuracy of the behavior models ($\log \pi_\beta$ on the y-axis) reduces during online fine-tuning, indicating that it is not fitting the new data well during online training. When the behavior models are inaccurate or unable to model new data well, constrained optimization becomes too conservative, resulting in limited improvement with fine-tuning. This analysis suggests that, in order to address our problem setting, we require an off-policy RL algorithm that constrains the policy to prevent offline instability and error accumulation, but not so conservatively that it prevents online fine-tuning due to imperfect behavior modeling. Our proposed

algorithm, which we discuss in the next section, accomplishes this by employing an *implicit* constraint, which does not require *any* explicit modeling of the behavior policy.

# 4 ADVANTAGE WEIGHTED ACTOR CRITIC: A SIMPLE ALGORITHM FOR FINE-TUNING FROM OFFLINE DATASETS

In this section, we will describe the advantage weighted actor-critic (AWAC) algorithm, which trains an off-policy critic and an actor with an *implicit* policy constraint. We will show AWAC mitigates the challenges outlined in Section 3. AWAC follows the design for actor-critic algorithms as described in Section 2, with a policy evaluation step to learn $Q^\pi$ and a policy improvement step to update $\pi$. AWAC uses off-policy temporal-difference learning to estimate $Q^\pi$ in the policy evaluation step, and a policy improvement update that is able to obtain the benefits of offline RL algorithms at training from prior datasets, while avoiding the overly conservative behavior described in Section 3.3. We describe the policy improvement step in AWAC below, and then summarize the entire algorithm.

Policy improvement for AWAC proceeds by learning a policy that maximizes the value of the critic learned in the policy evaluation step via TD bootstrapping. If done naively, this can lead to the issues described in Section 3.3, but we can avoid the challenges of bootstrap error accumulation by restricting the policy distribution to stay close to the data observed thus far during the actor update, while maximizing the value of the critic. At iteration $k$, AWAC therefore optimizes the policy to maximize the estimated Q-function $Q^{\pi_k}(\mathbf{s}, \mathbf{a})$ at every state, while constraining it to stay close to the actions observed in the data, similar to prior offline RL methods, though this constraint will be enforced differently. Note from the definition of the advantage in Section 2 that optimizing $Q^{\pi_k}(\mathbf{s}, \mathbf{a})$ is equivalent to optimizing $A^{\pi_k}(\mathbf{s}, \mathbf{a})$. We can therefore write this optimization as:

$$\pi_{k+1} = \arg\max_{\pi \in \Pi} \; \mathbb{E}_{\mathbf{a} \sim \pi(\cdot|\mathbf{s})}[A^{\pi_k}(\mathbf{s}, \mathbf{a})] \text{ s.t. } D_{\mathrm{KL}}(\pi(\cdot|\mathbf{s})||\pi_\beta(\cdot|\mathbf{s})) \le \epsilon. \tag{6}$$

As we saw in Section 3.2, enforcing the constraint by incorporating an explicit learned behavior model (Kumar et al., 2019; Fujimoto et al., 2019; Wu et al., 2020; Siegel et al., 2020) leads to poor fine-tuning performance. Instead, we enforce the constraint *implicitly*, without learning a behavior model. We first derive the solution to the constrained optimization in Equation 6 to obtain a non-parametric closed form for the actor. This solution is then projected onto the parametric policy class *without* any explicit behavior model. The analytic solution to Equation 6 can be obtained by enforcing the KKT conditions (Peters & Schaal, 2007; Peters et al., 2010; Peng et al., 2019). The Lagrangian is:

$$\mathcal{L}(\pi, \lambda) = \mathbb{E}_{\mathbf{a} \sim \pi(\cdot|\mathbf{s})}[A^{\pi_k}(\mathbf{s}, \mathbf{a})] + \lambda(\epsilon - D_{\mathrm{KL}}(\pi(\cdot|\mathbf{s})||\pi_\beta(\cdot|\mathbf{s}))), \tag{7}$$

and the closed form solution to this problem is $\pi^*(\mathbf{a}|\mathbf{s}) \propto \pi_\beta(\mathbf{a}|\mathbf{s}) \exp\left(\frac{1}{\lambda} A^{\pi_k}(\mathbf{s}, \mathbf{a})\right)$. When using function approximators, such as deep neural networks as we do, we need to project the non-parametric solution into our policy space. For a policy $\pi_\theta$ with parameters $\theta$, this can be done by minimizing the KL divergence of $\pi_\theta$ from the optimal non-parametric solution $\pi^*$ under the data distribution $\rho_{\pi_\beta}(\mathbf{s})$:

$$\arg\min_{\theta} \; \mathbb{E}_{\rho_{\pi_\beta}(\mathbf{s})}\left[D_{\mathrm{KL}}(\pi^*(\cdot|\mathbf{s})||\pi_\theta(\cdot|\mathbf{s}))\right] = \arg\min_{\theta} \; \mathbb{E}_{\rho_{\pi_\beta}(\mathbf{s})}\left[\mathbb{E}_{\pi^*(\cdot|\mathbf{s})}[-\log \pi_\theta(\cdot|\mathbf{s})]\right] \tag{8}$$

Note that the parametric policy could be projected with either direction of KL divergence. Choosing the reverse KL results in explicit penalty methods (Wu et al., 2020) that rely on evaluating the density of a learned behavior model. Instead, by using forward KL, we can compute the policy update by sampling directly from $\beta$:

$$\theta_{k+1} = \arg\max_{\theta} \; \mathbb{E}_{\mathbf{s}, \mathbf{a} \sim \beta}\left[\log \pi_\theta(\mathbf{a}|\mathbf{s}) \exp\left(\frac{1}{\lambda} A^{\pi_k}(\mathbf{s}, \mathbf{a})\right)\right]. \tag{9}$$

This actor update amounts to weighted maximum likelihood (i.e., supervised learning), where the targets are obtained by re-weighting the state-action pairs observed in the current dataset by the predicted advantages from the learned critic, *without* explicitly learning any parametric behavior model, simply sampling $(s, a)$ from the replay buffer $\beta$. See Appendix A.2 for a more detailed derivation and Appendix A.3 for specific implementation details.

**Avoiding explicit behavior modeling.** Note that the update in Equation 9 completely avoids any modeling of the previously observed data $\beta$ with a parametric model. By avoiding any explicit

learning of the behavior model AWAC is far less conservative than methods which fit a model $\hat{\pi}_\beta$ explicitly, and better incorporates new data during online fine-tuning, as seen from our results in Section 6. This derivation is related to AWR (Peng et al., 2019), with the main difference that AWAC uses an off-policy Q-function $Q^\pi$ to estimate the advantage, which greatly improves efficiency and even final performance (see results in Section 6.1). The update also resembles ABM-MPO, but ABM-MPO *does* require modeling the behavior policy which, as discussed in Section 3.3, can lead to poor fine-tuning. In Section 6.1, AWAC outperforms ABM-MPO on a range of challenging tasks.

**Policy evaluation.** During policy evaluation, we estimate the action-value $Q^\pi(\mathbf{s}, \mathbf{a})$ for the current policy $\pi$, as described in Section 2. We utilize a temporal difference learning scheme for policy evaluation (Haarnoja et al., 2018; Fujimoto et al., 2018), minimizing the Bellman error as described in Equation 2. This enables us to learn very efficiently from off-policy data. This is particularly important in our problem setting to effectively use the offline dataset, and allows us to significantly outperform alternatives using Monte-Carlo evaluation or TD($\lambda$) to estimate returns (Peng et al., 2019).

**Algorithm summary.** The full AWAC algorithm for offline RL with online fine-tuning is summarized in Algorithm 1. In a practical implementation, we can parameterize the actor and the critic by neural networks and perform SGD updates from Eqn. 9 and Eqn. 3. Specific details are provided in Appendix A.3. AWAC ensures data efficiency with off-policy critic estimation via bootstrapping, and avoids offline bootstrap error with a constrained actor update. By avoiding explicit modeling of the behavior policy, AWAC avoids overly conservative updates.

While AWAC is certainly quite related to several prior works, we note that there are key differences that make it particularly amenable to the problem setting we are considering - offline RL with online fine-tuning, that *none* of the other

---

**Algorithm 1** Advantage Weighted AC

1: Dataset $\mathcal{D} = \{(\mathbf{s}, \mathbf{a}, \mathbf{s}', r)_j\}$
2: Initialize buffer $\beta = \mathcal{D}$
3: Initialize $\pi_\theta, Q_\phi$
4: **for** iteration $i = 1, 2, ...$ **do**
5:     Sample batch $(\mathbf{s}, \mathbf{a}, \mathbf{s}', r) \sim \beta$
6:     Update $\phi$ according to Eqn. 3
7:     Update $\theta$ according to Eqn. 9
8:     **if** $i >$ num_offline_steps **then**
9:         $\tau_1, \ldots, \tau_K \sim p_{\pi_\theta}(\tau)$
10:        $\beta \leftarrow \beta \cup \{\tau_1, \ldots, \tau_K\}$
11:     **end if**
12: **end for**

---

methods are really able to tackle. As we show in our experimental analysis with direct comparisons to prior work, every one of the design decisions being made in this work are important for algorithm performance. As compared to AWR (Peng et al., 2019), AWAC uses TD bootstrapping for significantly more efficient and even asymptotically better performance. As compared to offline RL techniques like ABM (Siegel et al., 2020), MPO (Abdolmaleki et al., 2018), BEAR (Kumar et al., 2019) or BCQ (Fujimoto et al., 2019) this work is able to avoid the need for any behavior modeling, thereby enabling the *online* fine-tuning part of the problem much better. As shown in Fig 3, when these seemingly ablations are made to AWAC, the algorithm performs significantly worse.

## 5 RELATED WORK

Off-policy RL algorithms are designed to reuse off-policy data during training, and have been studied extensively (Konda & Tsitsiklis, 2000; Degris et al., 2012; Mnih et al., 2016; Haarnoja et al., 2018; Fujimoto et al., 2018; Bhatnagar et al., 2009; Peters & Schaal, 2008a; Zhang et al., 2019; Wawrzynski, 2009; Balduzzi & Ghifary, 2015). While standard off-policy methods are able to benefit from including data seen *during* a training run, as we show in Section 3.2 they struggle when training from previously collected offline data from other policies, due to error accumulation with distribution shift (Fujimoto et al., 2019; Kumar et al., 2019). Offline RL methods aim to address this issue, often by constraining the actor updates to avoid excessive deviation from the data distribution (Lange et al., 2012; Thomas & Brunskill, 2016; Hallak et al., 2015; 2016; Hallak & Mannor, 2017; Agarwal et al., 2019; Kumar et al., 2019; Fujimoto et al., 2019; Fakoor et al., 2019; Nachum et al., 2019; Siegel et al., 2020; Levine et al., 2020; Zhang et al., 2020). One class of these methods utilize importance sampling (Thomas & Brunskill, 2016; Zhang et al., 2020; Nachum et al., 2019; Degris et al., 2012; Jiang & Li, 2016; Hallak & Mannor, 2017). Another class of methods perform offline reinforcement learning via dynamic programming, with an explicit constraint to prevent deviation from the data distribution (Lange et al., 2012; Kumar et al., 2019; Fujimoto et al., 2019; Wu et al., 2020; Jaques et al., 2019). While these algorithms perform well in the purely offline settings, we show in Section 3.3 that such methods tend to be overly conservative, and therefore may not learn efficiently when fine-tuning with online data collection. In contrast, our algorithm AWAC is comparable to these algorithms for offline pre-training, but learns much more efficiently during subsequent fine-tuning.

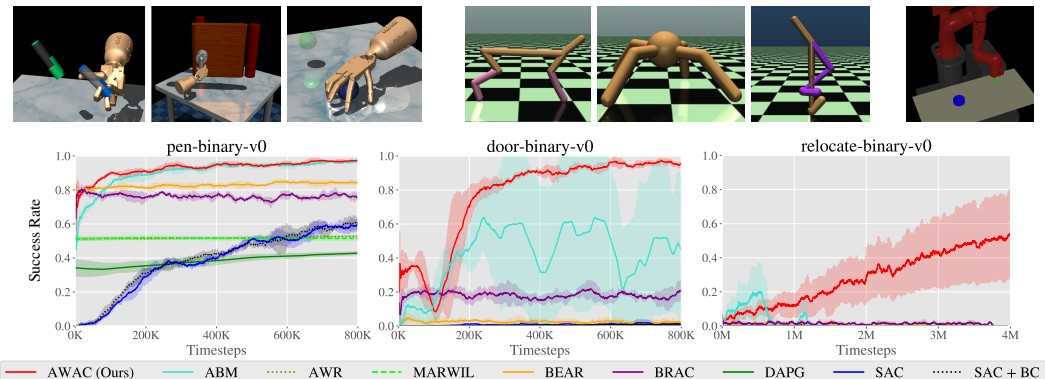

Figure 3: Comparative evaluation on the dexterous manipulation tasks. These tasks are difficult due to their high action dimensionality and reward sparsity. We see that AWAC is able to learn these tasks with little online data collection required (100K samples ≈ 16 minutes of equivalent real-world interaction time). Meanwhile, most prior methods are not able to solve the harder two tasks: door opening and object relocation.

Prior work has also considered the special case of learning from *demonstration* data. One class of algorithms initializes the policy via behavioral cloning from demonstrations, and then fine-tunes with reinforcement learning (Peters & Schaal, 2008b; Ijspeert et al., 2002; Theodorou et al., 2010; Kim et al., 2013; Rajeswaran et al., 2018; Gupta et al., 2019; Zhu et al., 2019). Most such methods use on-policy fine-tuning, which is less sample-efficient than off-policy methods that perform value function estimation. Other prior works have incorporated demonstration data into the replay buffer using off-policy RL methods (Večerík et al., 2017; Nair et al., 2017). We show in Section 3.2 that these strategies can result in a large dip in performance during online fine-tuning, due to the inability to pre-train an effective value function from offline data. In contrast, our work shows that using supervised learning style policy updates can allow for better bootstrapping from demonstrations as compared to Večerík et al. (2017) and Nair et al. (2017).

Our method builds on algorithms that implement a maximum likelihood objective for the actor, based on an expectation-maximization formulation of RL (Peters & Schaal, 2007; Neumann & Peters, 2008; Theodorou et al., 2010; Peters et al., 2010; Peng et al., 2019; Abdolmaleki et al., 2018; Wang et al., 2018). Most closely related to our method in this respect are the algorithms proposed by Peng et al. (2019) (AWR) and Siegel et al. (2020) (ABM). Unlike AWR, which estimates the value function of the *behavior* policy, $V^{\pi_\beta}$ via Monte-Carlo estimation or TD$-\lambda$, our algorithm estimates the Q-function of the *current* policy $Q^\pi$ via bootstrapping, enabling much more efficient learning, as shown in our experiments. Unlike ABM, our method does not require learning a separate function approximator to model the behavior policy $\pi_\beta$, and instead directly samples the dataset. As we discussed in Section 3.3, modeling $\pi_\beta$ can be a major challenge for online fine-tuning. While these distinctions may seem somewhat subtle, they are important and we show in our experiments that they result in a large difference in algorithm performance. Finally, our work goes beyond the analysis in prior work, by studying the issues associated with pre-training and fine-tuning in Section 3. Concurrently to our work, Wang et al. (2020) proposed critic regularized regression for offline RL, which uses off-policy Q-learning and an equivalent policy update. In contrast to this concurrent work, we specifically study the offline pretraining online fine-tuning problem, analyze why other methods are ineffective in this setting, and show that our approach achieves substantially better results.

## 6 Experimental Evaluation

In our experiments, we first compare our method against prior methods in the offline training and fine-tuning setting. We show that we can learn difficult, high-dimensional, sparse reward dexterous manipulation problems from human demonstrations and off-policy data. We then evaluate our method with suboptimal prior data generated by a random controller. Finally, we study why prior methods struggle in this setting by analyzing their performance on benchmark MuJoCo tasks, and conduct further experiments to understand where the difficulty lies (also shown in Section 3).

**6.1) Comparative Evaluation Learning From Prior Data.** We aim to study tasks representative of the difficulties of real-world robot learning, where offline learning and online fine-tuning are most

relevant. We begin our analysis with a set of challenging sparse reward dexterous manipulation tasks proposed by Rajeswaran et al. (2018). These tasks involve complex manipulation skills using a 28-DoF five-fingered hand in the MuJoCo simulator (Todorov et al., 2012) shown in Figure 3: in-hand rotation of a pen, opening a door by unlatching the handle, and picking up a sphere and relocating it to a target location. These environments exhibit many challenges: high dimensional action spaces, complex manipulation physics with many intermittent contacts, and randomized hand and object positions. The reward functions in these environments are binary 0-1 rewards for task completion. [2] Rajeswaran et al. (2018) provide 25 human demonstrations for each task, which are not fully optimal but do solve the task. Since this dataset is small, we generated another 500 trajectories of interaction data by constructing a behavioral cloned policy, and then sampling from this policy.

First, we compare our method on these dexterous manipulation tasks against prior methods for off-policy learning, offline learning, and bootstrapping from demonstrations. Specific implementation details are discussed in Appendix A.5. The results are shown in Fig. 3. Our method is able to leverage the prior data to quickly attain good performance, and the efficient off-policy actor-critic component of our approach fine-tunes much more quickly than demonstration augmented policy gradient (DAPG), the method proposed by Rajeswaran et al. (2018). For example, our method solves the pen task in 120K timesteps, the equivalent of just 20 minutes of online interaction. While the baseline comparisons and ablations are able to make some amount of progress on the pen task, alternative off-policy RL and offline RL algorithms are largely unable to solve the door and relocate task in the time-frame considered. We find that the design decisions to use off-policy critic estimation allow AWAC to significantly outperform AWR (Peng et al., 2019) while the implicit behavior modeling allows AWAC to significantly outperform ABM (Siegel et al., 2020), although ABM does make some progress. Rajeswaran et al. (2018) show that DAPG can solve variants of these tasks with more well-shaped rewards, but still requires considerably more samples.

Additionally, we evaluated all methods on the Gym MuJoCo locomotion benchmarks, similarly providing demonstrations as offline data. Due to space constraints, the results plots for these experiments are included in Appendix A.1. These tasks are substantially easier than the sparse reward manipulation tasks described above, and a number of prior methods also perform well. However, our method matches or exceeds the best prior method in all cases, whereas no other single prior method attains good performance on all of the tasks.

**6.2) Fine-Tuning from Random Policy Data.** An advantage of using off-policy RL for reinforcement learning is that we can also incorporate suboptimal data, rather than demonstrations. In this experiment, we evaluate on a simulated tabletop pushing environment with a Sawyer robot pictured in Fig 3 and described further in Appendix A.4. To study the potential to learn from suboptimal data, we use an off-policy dataset of 500 trajectories generated by a random process. The task is to push an object to a target location in a 40cm x 20cm goal space. The results are shown in Figure 4. We see that while many methods begin at the same initial performance, AWAC learns the fastest online and is actually able to make use of the offline dataset effectively.

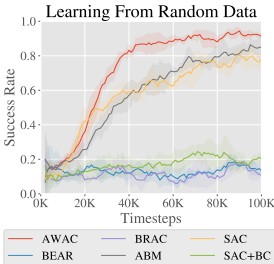

Figure 4: Comparison of fine-tuning from an initial dataset of suboptimal data on a Sawyer robot pushing task.

## 7  DISCUSSION AND FUTURE WORK

We have discussed in detail the challenges existing RL methods face when fine-tuning from prior datasets, and proposed an algorithm, AWAC, that is effective in this setting. The key insight in AWAC is that enforcing a policy update constraint implicitly on actor-critic methods results in a stable learning algorithm amenable for off-policy learning. With an informative action-value estimate, the policy is weighted towards high-advantage actions in the data, resulting in policy improvement without conservative updates. A direction of future work we plan to pursue is applying AWAC to solve difficult robotic tasks in the real world. More than just speeding up individual runs, incorporating prior data into the learning process enables continuously accumulating data by saving environment interactions of the robot - for instance, runs of RL with varying hyperparameters. We hope that this enables a wider array of robotic applications than previously possible.

---

[2]Rajeswaran et al. (2018) use a combination of task completion factors as the sparse reward. For instance, in the door task, the sparse reward as a function of the door position $d$ was $r = 10\mathbb{1}_{d>1.35} + 8\mathbb{1}_{d>1.0} + 2\mathbb{1}_{d>1.2} - 0.1||d - 1.57||_2$. We only use the success measure $r = \mathbb{1}_{d>1.4}$, which is substantially more difficult.

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

# A APPENDIX

## A.1 GYM BENCHMARK RESULTS FROM PRIOR DATA

In this section, we provide a comparative evaluation on MuJoCo benchmark tasks for analysis. These tasks are simpler, with dense rewards and relatively lower action and observation dimensionality. Thus, many prior methods can make good progress on these tasks. These experiments allow us to understand more precisely which design decisions are crucial. For each task, we collect 15 demonstration trajectories using a pre-trained expert on each task, and 100 trajectories of off-policy data by rolling out a behavioral cloned policy trained on the demonstrations. The same data is made available to all methods. The results are presented in Figure 5. AWAC is consistently the best or on par with the best-performing method. No other single method consistently attains the best results – on HalfCheetah, SAC + BC and BRAC are competitive, while on Ant-v2 ABM is competitive with AWAC. We summarize the results according to the challenges in Section 3.

**Data efficiency.** The three methods that do not estimate $Q^\pi$ are DAPG (Abdolmaleki et al., 2018), AWR (Peng et al., 2019), and MARWIL (Wang et al., 2018). Across all three tasks, we see that these methods are somewhat worse offline than the best performing offline methods, and exhibit steady but very slow improvement during fine-tuning. In robotics, data efficiency is vital, so these algorithms are not good candidates for practical real-world applications.

**Bootstrap error in offline learning.** For SAC (Haarnoja et al., 2018), across all three tasks, we see that the offline performance at epoch 0 is generally poor. Due to the data in the replay buffer, SAC with prior data does learn faster than from scratch, but AWAC is faster to solve the tasks in general. SAC with additional data in the replay buffer is similar to the approach proposed by Večerík et al. (2017). SAC+BC reproduces Nair et al. (2018) but uses SAC instead of DDPG (Lillicrap et al., 2016) as the underlying RL algorithm. We find that these algorithms exhibit a characteristic dip at the start of learning. Although this dip is only present in the early part of the learning curve, a poor initial policy and lack of steady policy improvement can be a safety concern and a significant hindrance in real-world applications. Moreover, recall that in the more difficult dextrous manipulation tasks, these algorithms do not show any significant learning.

**Conservative online learning.** Finally, we consider conservative offline algorithms: ABM (Siegel et al., 2020), BEAR (Kumar et al., 2019), and BRAC (Wu et al., 2020). We found that BRAC performs similarly to SAC for working hyperparameters. BEAR trains well offline – on Ant and Walker2d, BEAR significantly outperforms prior methods before online experience. However, online improvement is slow for BEAR and the final performance across all three tasks is much lower than AWAC. The closest in performance to our method is ABM, which is comparable on Ant-v2, but much slower on other domains.

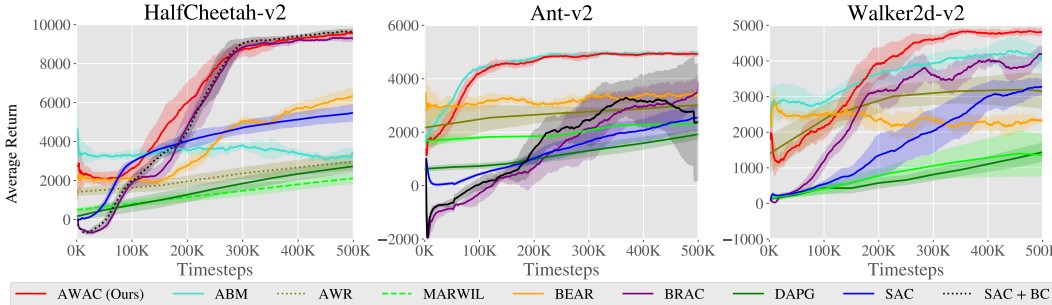

Figure 5: Comparison of our method and prior methods on standard MuJoCo benchmark tasks. These tasks are much easier than the dexterous manipulation tasks, and allow us to better inspect the performance of methods in the setting of offline pretraining followed by online fine-tuning. SAC+BC and BRAC perform on par with our method on the HalfCheetah task, and ABM performs on par with our method on the Ant task, while our method outperforms all others on the Walker2D task. Our method matches or exceeds the best prior method in all cases, whereas no other single prior method attains good performance on all of the tasks.

A.2   ALGORITHM DERIVATION DETAILS

The full optimization problem we solve, given the previous off-policy advantage estimate $A^{\pi_k}$ and buffer distribution $\pi_\beta$, is given below:

$$\pi_{k+1} = \underset{\pi \in \Pi}{\arg\max} \, \mathbb{E}_{\mathbf{a} \sim \pi(\cdot|\mathbf{s})}[A^{\pi_k}(\mathbf{s}, \mathbf{a})] \tag{10}$$

$$\text{s.t. } D_{\mathrm{KL}}(\pi(\cdot|\mathbf{s})||\pi_\beta(\cdot|\mathbf{s})) \leq \epsilon \tag{11}$$

$$\int_\mathbf{a} \pi(\mathbf{a}|\mathbf{s})d\mathbf{a} = 1. \tag{12}$$

Our derivation follows Peters et al. (2010) and Peng et al. (2019). The analytic solution for the constrained optimization problem above can be obtained by enforcing the KKT conditions. The Lagrangian is:

$$\mathcal{L}(\pi, \lambda, \alpha) = \mathbb{E}_{\mathbf{a} \sim \pi(\cdot|\mathbf{s})}[A^{\pi_k}(\mathbf{s}, \mathbf{a})] + \lambda(\epsilon - D_{\mathrm{KL}}(\pi(\cdot|\mathbf{s})||\pi_\beta(\cdot|\mathbf{s}))) + \alpha(1 - \int_\mathbf{a} \pi(\mathbf{a}|\mathbf{s})d\mathbf{a}). \tag{13}$$

Differentiating with respect to $\pi$ gives:

$$\frac{\partial \mathcal{L}}{\partial \pi} = A^{\pi_k}(\mathbf{s}, \mathbf{a}) - \lambda \log \pi_\beta(\mathbf{a}|\mathbf{s}) + \lambda \log \pi(\mathbf{a}|\mathbf{s}) + \lambda - \alpha. \tag{14}$$

Setting $\frac{\partial \mathcal{L}}{\partial \pi}$ to zero and solving for $\pi$ gives the closed form solution to this problem:

$$\pi^*(\mathbf{a}|\mathbf{s}) = \frac{1}{Z(\mathbf{s})}\pi_\beta(\mathbf{a}|\mathbf{s})\exp\left(\frac{1}{\lambda}A^{\pi_k}(\mathbf{s}, \mathbf{a})\right), \tag{15}$$

Next, we project the solution into the space of parametric policies. For a policy $\pi_\theta$ with parameters $\theta$, this can be done by minimizing the KL divergence of $\pi_\theta$ from the optimal non-parametric solution $\pi^*$ under the data distribution $\rho_{\pi_\beta}(\mathbf{s})$:

$$\underset{\theta}{\arg\min} \, \underset{\rho_{\pi_\beta}(\mathbf{s})}{\mathbb{E}} \left[D_{\mathrm{KL}}(\pi^*(\cdot|\mathbf{s})||\pi_\theta(\cdot|\mathbf{s}))\right] = \underset{\theta}{\arg\min} \, \underset{\rho_{\pi_\beta}(\mathbf{s})}{\mathbb{E}} \left[\underset{\pi^*(\cdot|\mathbf{s})}{\mathbb{E}}[-\log \pi_\theta(\cdot|\mathbf{s})]\right] \tag{16}$$

Note that in the projection step, the parametric policy could be projected with either direction of KL divergence. However, choosing the reverse KL direction has a key advantage: it allows us to optimize $\theta$ as a maximum likelihood problem with an expectation over data $s, a \sim \beta$, rather than sampling actions from the policy that may be out of distribution for the Q function. In our experiments we show that this decision is vital for stable off-policy learning.

Furthermore, assume discrete policies with a minimum probably density of $\pi_\theta \geq \alpha_\theta$. Then the upper bound:

$$D_{\mathrm{KL}}(\pi^*||\pi_\theta) \leq \frac{2}{\alpha_\theta}D_{\mathrm{TV}}(\pi^*, \pi_\theta)^2 \tag{17}$$

$$\leq \frac{1}{\alpha_\theta}D_{\mathrm{KL}}(\pi_\theta||\pi^*) \tag{18}$$

holds by the Pinsker's inequality, where $D_{\mathrm{TV}}$ denotes the total variation distance between distributions. Thus minimizing the reverse KL also bounds the forward KL. Note that we can control the minimum $\alpha$ if desired by applying Laplace smoothing to the policy.

A.3   IMPLEMENTATION DETAILS

We implement the algorithm building on top of twin soft actor-critic (Haarnoja et al., 2018), which incorporates the twin Q-function architecture from twin delayed deep deterministic policy gradient (TD3) from Fujimoto et al. (2018). All off-policy algorithm comparisons (SAC, BRAC, MPO, ABM, BEAR) are implemented from the same skeleton. The base hyperparameters are given in Table 2. The policy update is replaced with:

$$\theta_{k+1} = \underset{\theta}{\arg\max} \, \underset{\mathbf{s}, \mathbf{a} \sim \beta}{\mathbb{E}} \left[\log \pi_\theta(\mathbf{a}|\mathbf{s})\frac{1}{Z(\mathbf{s})}\exp\left(\frac{1}{\lambda}A^{\pi_k}(\mathbf{s}, \mathbf{a})\right)\right]. \tag{19}$$

Similar to advantage weight regression (Peng et al., 2019) and other prior work (Neumann & Peters, 2008; Wang et al., 2018; Siegel et al., 2020), we disregard the per-state normalizing constant $Z(\mathbf{s}) = \int_{\mathbf{a}} \pi_\theta(\mathbf{a}|\mathbf{s}) \exp\left(\frac{1}{\lambda} A^{\pi_k}(\mathbf{s}, \mathbf{a})\right) d\mathbf{a} = \mathbb{E}_{\mathbf{a} \sim \pi_\theta(\cdot|\mathbf{s})}[A^{\pi_k}(\mathbf{s}, \mathbf{a})]$. We did experiment with estimating this expectation per batch element with $K = 10$ samples, but found that this generally made performance worse, perhaps because errors in the estimation of $Z(\mathbf{s})$ caused more harm than the benefit the method derived from estimating this value. We report success rate results for variants of our method with and without $Z(\mathbf{s})$ estimation in Table 1.

| Env | Use $Z(\mathbf{s})$ | Omit $Z(\mathbf{s})$ |
|---|---|---|
| pen | 84% | 98% |
| door | 0% | 95% |
| relocate | 0% | 54% |

Table 1: Success rates after online fine-tuning (after 800K steps for pen, door and 4M steps for relocate) using AWAC with and without $Z(\mathbf{s})$ weight. These results show that although we can estimate $Z(\mathbf{s})$, weighting by $Z(\mathbf{s})$ actually results in worse performance.

While prior work (Neumann & Peters, 2008; Wang et al., 2018; Peng et al., 2019) has generally ignored the omission of $Z(\mathbf{s})$ without any specific justification, it is possible to bound this value both above and below using the Cauchy-Schwarz and reverse Cauchy-Schwarz (Polya-Szego) inequalities, as follows. Let $f(\mathbf{a}) = \pi(\mathbf{a}|\mathbf{s})$ and $g(\mathbf{a}) = \exp(A(\mathbf{s}, \mathbf{a})/\lambda)$. Note $f(\mathbf{a}) > 0$ for stochastic policies and $g(\mathbf{a}) > 0$. By Cauchy-Schwarz, $Z(s) = \int_{\mathbf{a}} f(\mathbf{a})g(\mathbf{a})d\mathbf{a} \leq \sqrt{\int_{\mathbf{a}} f(\mathbf{a})^2 d\mathbf{a} \int_{\mathbf{a}} g(\mathbf{a})^2 d\mathbf{a}} = C_1$. To apply Polya-Szego, let $m_f$ and $m_g$ be the minimum of $f$ and $g$ respectively and $M_f, M_g$ be the maximum. Then $Z(\mathbf{s}) \geq 2(\sqrt{\frac{M_f M_g}{m_f m_g}} + \frac{m_f m_g}{M_f M_g})^{-1} C_1 = C_2$. We therefore have $C_1 \leq Z(\mathbf{s}) \leq C_2$, though the bounds are generally not tight.

A further, more intuitive argument for why omitting $Z(\mathbf{s})$ may be harmless in practice comes from observing that this normalizing factor only affects the relative weight of different *states* in the training objective, not different actions. The state distribution in $\beta$ already differs from the distribution over states that will be visited by $\pi_\theta$, and therefore preserving this state distribution is likely to be of limited utility to downstream policy performance. Indeed, we would expect that sufficiently expressive policies would be less affected by small to moderate variability in the state weights. On the other hand, inaccurate estimates of $Z(\mathbf{s})$ may throw off the training objective by increasing variance, similar to the effect of degenerate importance weights.

The Lagrange multiplier $\lambda$ is treated as a hyperparameter in our method. In this work we use $\lambda = 0.3$ for the manipulation environments and $\lambda = 1.0$ for the MuJoCo benchmark environments. One could adaptively learn $\lambda$ with a dual gradient descent procedure, but this would require access to $\pi_\beta$.

As rewards for the dextrous manipulation environments are non-positive, we clamp the Q value for these experiments to be at most zero. We find this stabilizes training slightly.

## A.4 ENVIRONMENT-SPECIFIC DETAILS

We evaluate our method on three domains: dexterous manipulation environments, Sawyer manipulation environments, and MuJoCo benchmark environments. In the following sections we describe specific details.

### A.4.1 DEXTEROUS MANIPULATION ENVIRONMENTS

These environments are modified from those proposed by Rajeswaran et al. (2018).

**pen-binary-v0.** The task is to spin a pen into a given orientation. The action dimension is 24 and the observation dimension is 45. Let the position and orientation of the pen be denoted by $x_p$ and $x_o$ respectively, and the desired position and orientation be denoted by $d_p$ and $d_o$ respectively. The reward function is $r = \mathbb{1}_{|x_p - d_p| \leq 0.075} \mathbb{1}_{|x_o \cdot d_o| \leq 0.95}$ - 1. In Rajeswaran et al. (2018), the episode was terminated when the pen fell out of the hand; we did not include this early termination condition.

**door-binary-v0.** The task is to open a door, which requires first twisting a latch. The action dimension is 28 and the observation dimension is 39. Let $d$ denote the angle of the door. The reward function is $r = \mathbb{1}_{d > 1.4}$ - 1.

| Hyper-parameter | Value |
|---|---|
| Training Batches Per Timestep | 1 |
| Exploration Noise | None (stochastic policy) |
| RL Batch Size | 1024 |
| Discount Factor | 0.99 |
| Reward Scaling | 1 |
| Replay Buffer Size | 1000000 |
| Number of pretraining steps | 25000 |
| Policy Hidden Sizes | $[256, 256, 256, 256]$ |
| Policy Hidden Activation | ReLU |
| Policy Weight Decay | $10^{-4}$ |
| Policy Learning Rate | $3 \times 10^{-4}$ |
| Q Hidden Sizes | $[256, 256, 256, 256]$ |
| Q Hidden Activation | ReLU |
| Q Weight Decay | 0 |
| Q Learning Rate | $3 \times 10^{-4}$ |
| Target Network $\tau$ | $5 \times 10^{-3}$ |

Table 2: Hyper-parameters used for RL experiments.

**relocate-binary-v0.** The task is to relocate an object to a goal location. The action dimension is 30 and the observation dimension is 39. Let $x_p$ denote the object position and $d_p$ denote the desired position. The reward is $r = \mathbb{1}_{|x_p - d_p| \leq 0.1}$ - 1.

### A.4.2 SAWYER MANIPULATION ENVIRONMENT

**SawyerPush-v0.** This environment is included in the Multiworld library. The task is to push a puck to a goal position in a 40cm x 20cm, and the reward function is the negative distance between the puck and goal position. When using this environment, we use hindsight experience replay for goal-conditioned reinforcement learning. The random dataset for prior data was collected by rolling out an Ornstein-Uhlenbeck process with $\theta = 0.15$ and $\sigma = 0.3$.

### A.4.3 OFF-POLICY DATA PERFORMANCE

The performances of the expert data, behavior cloning (BC) on the expert data (1), and BC on the combined expert+BC data (2) are included in Table 3. For Gym benchmarks we report average return, and expert data is collected by a trained SAC policy. For dextrous manipulation tasks we report the success rate, and the expert data consists of human demonstrations provided by Rajeswaran et al. (2018).

| Env | Expert | BC (1) | BC (2) |
|---|---|---|---|
| cheetah | 9962 | 2507 | 4524 |
| walker | 5062 | 2040 | 1701 |
| ant | 5207 | 687 | 1704 |
| pen | 1 | 0.73 | 0.76 |
| door | 1 | 0.10 | 0.00 |
| relocate | 1 | 0.02 | 0.01 |

Table 3: Performance of the off-policy data for each environment. BC (1) indicates BC on the expert data, while BC (2) indicates BC on the combined expert+BC data used as off-policy data for pretraining.

### A.5 BASELINE IMPLEMENTATION DETAILS

We used public implementations of prior methods (DAPG, AWR) when available. We implemented the remaining algorithms in our framework, which also allows us to understand the effects of changing individual components of the method. In the section, we describe the implementation details. The full overview of algorithms is given in Figure 6.

| Name | $\hat{Q}$ | Policy Objective | $\hat{\pi}_\beta$? | Constraint |
|------|-----------|------------------|--------------------|------------|
| SAC | $Q^\pi$ | $D_{\mathrm{KL}}(\pi_\theta \| \bar{Q})$ | No | None |
| SAC + BC | $Q^\pi$ | Mixed | No | None |
| BCQ | $Q^\pi$ | $D_{\mathrm{KL}}(\pi_\theta \| \bar{Q})$ | Yes | Support ($\ell^\infty$) |
| BEAR | $Q^\pi$ | $D_{\mathrm{KL}}(\pi_\theta \| \bar{Q})$ | Yes | Support (MMD) |
| AWR | $Q^\beta$ | $D_{\mathrm{KL}}(\bar{Q} \| \pi_\theta)$ | No | Implicit |
| MPO | $Q^\pi$ | $D_{\mathrm{KL}}(\bar{Q} \| \pi_\theta)$ | Yes* | Prior |
| ABM-MPO | $Q^\pi$ | $D_{\mathrm{KL}}(\bar{Q} \| \pi_\theta)$ | Yes | Learned Prior |
| DAPG | - | $J(\pi_\theta)$ | No | None |
| BRAC | $Q^\pi$ | $D_{\mathrm{KL}}(\pi_\theta \| \bar{Q})$ | Yes | Explicit KL penalty |
| AWAC (Ours) | $Q^\pi$ | $D_{\mathrm{KL}}(\bar{Q} \| \pi_\theta)$ | No | Implicit |

Figure 6: Comparison of prior algorithms that can incorporate prior datasets. See section A.5 for specific implementation details. We argue that avoiding estimating $\hat{\pi}_\beta$ (i.e., $\hat{\pi}_\beta$ is "No") is important when learning with complex datasets that include experience from multiple policies, as in the case of online fine-tuning, and maintaining a constraint of some sort is essential for offline training. At the same time, sample-efficient learning requires using $Q^\pi$ for the critic. Our algorithm is the only one that fulfills all of these requirements.

**Behavior Cloning (BC).** This method learns a policy with supervised learning on demonstration data.

**Soft Actor Critic (SAC).** Using the soft actor critic algorithm from (Haarnoja et al., 2018), we follow the exact same procedure as our method in order to incorporate prior data, initializing the policy with behavior cloning on demonstrations and adding all prior data to the replay buffer.

**Behavior Regularized Actor Critic (BRAC).** We implement BRAC as described in (Wu et al., 2020) by adding policy regularization $\log(\pi_\beta(a|s))$ where $\pi_\beta$ is a behavior policy trained with supervised learning on the replay buffer. We add all prior data to the replay buffer before online training.

**Advantage Weighted Regression (AWR).** Using the advantage weighted regression algorithm from (Peng et al., 2019), we add all prior data to the replay buffer before online training. We use the implementation provided by Peng et al. (2019), with the key difference from our method being that AWR uses TD($\lambda$) on the replay buffer for policy evaluation.

**Monotonic Advantage Re-Weighted Imitation Learning (MARWIL).** Monotonic advantage re-weighted imitation learning was proposed by Wang et al. (2018) for offline imitation learning. MARWIL was not demonstrated in online RL settings, but we evaluate it for offline pretraining followed by online fine-tuning as we do other offline algorithms. Although derived differently, MARWIL and AWR are similar algorithms and only differ in value estimation: MARWIL uses the on-policy single-path advantage estimate $A(s,a) = Q^{\pi_\beta}(s,a) - V^{\pi_\beta}(s)$ instead of TD($\lambda$) as in AWR. Thus, we implement MARWIL by modifying the implementation of AWR.

**Maximum a Posteriori Policy Optimization (MPO).** We evaluate the MPO algorithm presented by Abdolmaleki et al. (2018). Due to a public implementation being unavailable, we modify our algorithm to be as close to MPO as possible. In particular, we change the policy update in Advantage Weighted Actor Critic to be:

$$\theta_i \longleftarrow \arg\max_{\theta_i} \quad \mathbb{E}_{s \sim \mathcal{D}, a \sim \pi(a|s)} \left[ \log \pi_{\theta_i}(a|s) \exp(\frac{1}{\beta} Q^{\pi_\beta}(s,a)) \right]. \tag{20}$$

Note that in MPO, actions for the update are sampled from the policy and the Q-function is used instead of advantage for weights. We failed to see offline or online improvement with this implementation in most environments, so we omit this comparison in favor of ABM.

**Advantage-Weighted Behavior Model (ABM).** We evaluate ABM, the method developed in Siegel et al. (2020). As with MPO, we modify our method to implement ABM, as there is no public

implementation of the method. ABM first trains an advantage model $\pi_{\theta_{\text{abm}}}(a|s)$:

$$\theta_{\text{abm}} = \arg\max_{\theta_i} \ \mathbb{E}_{\tau \sim \mathcal{D}} \left[ \sum_{t=1}^{|\tau|} \log \pi_{\theta_{\text{abm}}}(a_t|s_t) f(R(\tau_{t:N}) - \hat{V}(s)) \right]. \tag{21}$$

where $f$ is an increasing non-negative function, chosen to be $f = 1_+$. In place of an advantage computed by empirical returns $R(\tau_{t:N}) - \hat{V}(s)$ we use the advantage estimate computed per transition by the $Q$ value $Q(s, a) - V(s)$. This is favorable for running ABM online, as computing $R(\tau_{t:N}) - \hat{V}(s)$ is similar to AWR, which shows slow online improvement. We then use the policy update:

$$\theta_i \longleftarrow \arg\max_{\theta_i} \ \mathbb{E}_{s \sim \mathcal{D}, a \sim \pi_{\text{abm}}(a|s)} \left[ \log \pi_{\theta_i}(a|s) \exp \left( \frac{1}{\lambda} (Q^{\pi_i}(s, a) - V^{\pi_i}(s)) \right) \right]. \tag{22}$$

Additionally, for this method, actions for the update are sampled from a behavior policy trained to match the replay buffer and the value function is computed as $V^{\pi}(s) = Q^{\pi}(s, a)$ s.t. $a \sim \pi$.

**Demonstration Augmented Policy Gradient (DAPG).** We directly utilize the code provided in (Rajeswaran et al., 2018) to compare against our method. Since DAPG is an on-policy method, we only provide the demonstration data to the DAPG code to bootstrap the initial policy from.

**Bootstrapping Error Accumulation Reduction (BEAR).** We utilize the implementation of BEAR provided in rlkit. We provide the demonstration and off-policy data to the method together. Since the original method only involved training offline, we modify the algorithm to include an online training phase. In general we found that the MMD constraint in the method was too conservative. As a result, in order to obtain the results displayed in our paper, we swept the MMD threshold value and chose the one with the best final performance after offline training with offline fine-tuning.

## A.6 Extra Baseline Comparisons (CQL, AlgaeDICE)

In this section, we add comparisons to constrained Q-learning (CQL) (Kumar et al., 2020) and AlgaeDICE (Nachum et al., 2019). For CQL, we use the authors' implementation, modified for additionally online-finetuning instead of only offline training. For AlgaeDICE, we use the publicly available implementation, modified to load prior data and perform 25K pretraining steps before online RL. The results are presented in Figure 7.

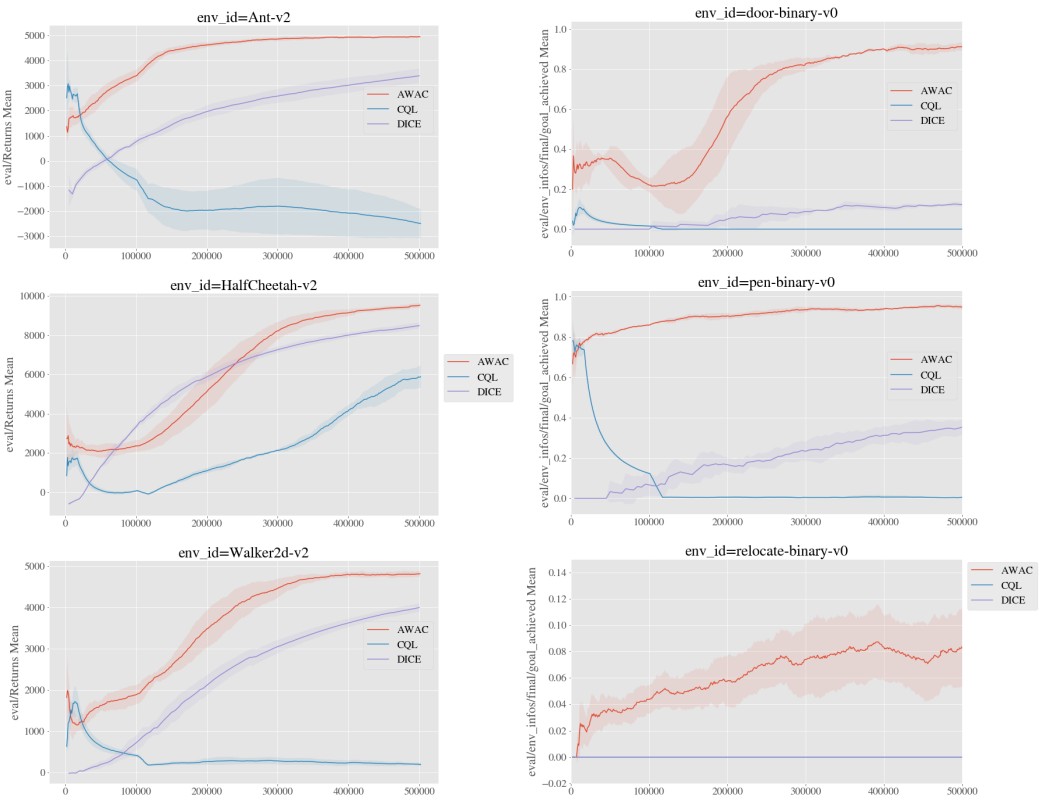

Figure 7: Comparison of our method (AWAC) with CQL and AlgaeDICE. CQL and AWAC perform similarly offline, but CQL does not improve when fine-tuning online. AlgaeDICE does not perform well for offline pretraining.

## A.7 ONLINE FINE-TUNING FROM D4RL

In this experiment, we evaluate the performance of varied data quality (random, medium, medium-expert, and expert) datasets included in D4RL (Fu et al., 2020), a dataset intended for offline RL. The results are obtained by first by training offline and then fine-tuning online on each setting for 500,000 additional steps. The performance of BEAR (Kumar et al., 2019) is attached as reference. We attempted to fine-tune BEAR online using the same protocol as AWAC but the performance did not improve and often decreased; thus we report the offline performance. All performances are scaled to 0 to 100, where 0 is the average returns of a random policy and 100 is the average returns of an expert policy (obtained by training online with SAC), as is standard for D4RL.

The results are presented in Figure 8. First, we observe that AWAC (offline) is competitive with BEAR, a commonly used offline RL algorithm. Then, AWAC is able to make progress in solving the tasks with online fine-tuning, even when initialized from random data or "medium" quality data, as shown by the performance of AWAC (online). In almost all settings, AWAC (online) is the best performing or tied with BEAR. In four of the six lower quality (random or medium) data settings, AWAC (online) is significantly better than BEAR; it is reasonable that AWAC excels in the lower-quality data regime because there is more room for online improvement, while both offline RL methods often start at high performance when initialized from higher-quality data.

|  |  | AWAC (offline) | AWAC (online) | BEAR |
|---|---|---|---|---|
| HalfCheetah | random | 2.2 | **52.9** | 25.5 |
|  | medium | 37.4 | 41.1 | 38.6 |
|  | medium-expert | 36.8 | 41.0 | **51.7** |
|  | expert | 78.5 | 105.6 | 108.2 |
| Hopper | random | 9.6 | **62.8** | 9.5 |
|  | medium | 72.0 | **91.0** | 47.6 |
|  | medium-expert | 80.9 | **111.9** | 4.0 |
|  | expert | 85.2 | 111.8 | 110.3 |
| Walker2D | random | 5.1 | 11.7 | 6.7 |
|  | medium | 30.1 | **79.1** | 33.2 |
|  | medium-expert | 42.7 | **78.3** | 10.8 |
|  | expert | 57.0 | 103.0 | 106.1 |

Figure 8: Comparison of our method (AWAC) fine-tuning on varying data quality datasets in D4RL (Fu et al., 2020). AWAC is able to improve its offline performance by further fine-tuning online.

