# OpenReview forum: "AWAC: Accelerating Online Reinforcement Learning with Offline Datasets"
_ICLR.cc/2021/Conference — Reject_

### Official Review · AnonReviewer4 · 2020-10-28
**Incremental but practically interesting approach**

**Rating:** 6
**Confidence:** 4

**Review:**

The paper presents an improved actor-critic algorithm for learning from offline data while requiring minimum interactions during online learning. The approach implicitly updates a low-variance policy without the need to estimate the sampling policy. The empirical study shows that AWAC leads to a faster convergence rate compared to the competitors.

Overall, the paper provides an incremental contribution over the existing works for off-policy RL problems. However, the proposed algorithm demonstrates some practical benefits in simulated scenarios and is analyzes from different perspectives. Hence, I vote for accepting, given the following concerns are addressed in the rebuttal period.

Pros:
- Interesting adaption of algorithms to effectively use the offline data in online RL scenarios which is independent from the sampling policy
- Ensuring the policy to stay close to the observed data without directly estimating the sampling policy
- Nicely motivated with helpful examples
- Clarifies differences with the related work

Comments:
- The presentation of the paper could be improved. There are some inconsistencies in the text. At the beginning the approach is introduced to be based on dynamic programming and then turns into temporal difference learning, which is misleading. In addition, offline pre-training and online fine-tuning is not quite accurate as the algorithm 1 is using the mix of data in one phase which does not involve pre-training, or is it actually in two phases?

- The paper seems to be an incremental improvement/combination of several recent works mentioned in the paper.

- The experiments are all on simulated environments. It would be interesting to see how the approach performs on real-world applications, and with some other evaluation metric than convergence rate.

- Demonstration data is not always as complete as mentioned in the paper. Expert demonstrations that are used for imitation learning, normally lack the reward values and in the form of (s,a) pairs only.


Minor:
-Typo in section 3.2: is policy is pre-trained
- Plot 4 in Fig. 2 not addressed in section 3.3

---

> ### Author Response · Authors · 2020-11-14
> **Initial response to R4**
>
> Thank you for the review.
>
> > Algorithm 1
>
> 1. For i <= num_offline_steps, no additional data is collected in the environment, so this is what we call the offline pre-training phase. 2. In the second phase (online fine-tuning), we collect online data. For training updates we use all available data for the algorithm: both the offline and online data together. Does that clear up the concern?
>
> > Demonstration data lacks rewards
>
> This is true of imitation learning methods; the prior methods (DAPG [1], SACfD [2], and SAC+BC [3]) that bootstrap from demonstrations that we compare to also have the same assumption however of having access to rewards in the demonstration data. Additionally, we show that we can also take advantage of arbitrary off-policy data.
>
> > incremental improvement/combination of several recent works mentioned in the paper.
>
> As we also discuss in the shared response, while the proposed algorithm shares some similarities with prior works, it makes key design decisions in the choice of how to enforce the actor update and the critic update which actually allow the algorithm to pretrain offline and finetune effectively online, without any explicit behavior modeling. While these design decisions may seem somewhat subtle, we provide a careful analysis in Section 3, which shows how these decisions make the difference between the algorithm being able to solve challenging dexterous manipulation tasks or not.
>
> [1] Aravind Rajeswaran, Vikash Kumar, Abhishek Gupta, John Schulman, Emanuel Todorov, and Sergey Levine.   Learning Complex Dexterous Manipulation with Deep Reinforcement Learning and Demonstrations. In Robotics: Science and Systems, 2018.
>
> [2] Matej Vecerík, Todd Hester, Jonathan Scholz, Fumin Wang, Olivier Pietquin, Bilal Piot, Nicolas Heess, Thomas Rothörl, Thomas Lampe, and Martin Riedmiller. Leveraging Demonstrations for Deep Reinforcement Learning on Robotics Problems with Sparse Rewards. CoRR, abs/1707.0, 2017.
>
> [3] Ashvin Nair, Bob Mcgrew, Marcin Andrychowicz, Wojciech Zaremba, and Pieter Abbeel.  Over-coming Exploration in Reinforcement Learning with Demonstrations.   In IEEE International Conference on Robotics and Automation (ICRA), 2018.

---

### Official Review · AnonReviewer1 · 2020-10-28
**Lack of novelty**

**Rating:** 3
**Confidence:** 4

**Review:**

This paper studies challenges of offline RL with online fine-tuning and proposes an off-policy actor-critic method to address these challenges. The proposed method uses a supervised learning style to update the model parameters and avoids the behavior model estimation.  Empirical results show that the proposed method provides rapid learning with prior demonstration data and online experience.

I have a major concern about the novelty of this paper. The major component of the proposed AWAC method, updating the model parameter using supervised learning, is exactly the same as AWR (Peng et. al., 2019). One minor difference seems that AWAC uses an off-policy policy evaluation, but this contribution is very marginal.

This paper provides an analysis on challenges of combining offline RL with online improvement, which motivates this paper. However, most of the discussed challenges are quite well-known. For example, one major challenge is to estimate the behavior model in offline data. This paper does not discuss techniques/methods that address this challenge, like DualDICE (Nachum et al., 2019) and CQL (Kumaret al., 2020). A comparison to these methods is highly recommended. In addition, this paper may also include additional model-based offline RL methods, like MOPO.

---

> ### Author Response · Authors · 2020-11-14
> **Initial response to R1**
>
> Thank you for the review. We have addressed the contribution vs. AWR in the common thread: AWAC achieves significantly better offline performance and subsequent online fine-tuning performance than AWR in all of the tasks evaluated (Figures 3, 4, 5).
>
> We have now evaluated CQL to add as a comparison. See this link (https://imgur.com/a/t8zcMZf) for results on the six environments. CQL performs well offline, comparable to our method, but does not fine-tune well online on five of the six tasks. As CQL is designed for offline RL, this is not too surprising. Additionally note that the contributions of CQL and AWAC are somewhat orthogonal, in that the min-Q loss in CQL could be added for value estimation in AWAC. DualDICE proposes a method for off-policy correction of the (s, a) data distribution for off-policy evaluation, and this distribution correction was incorporated into an RL algorithm in AlgaeDICE. We will add a comparison with AlgaeDICE in the following revision.

---

> ### Author Response · Authors · 2020-11-21
> **Comparisons against CQL and DICE**
>
> We have now added the comparison to CQL and AlgaeDICE to the paper as appendix section A.6. We attach it here for convenience: https://imgur.com/a/r7dIlBF. On all six environments, AWAC outperforms CQL and AlgaeDICE. CQL and AWAC perform similarly offline, but CQL does not improve when fine-tuning online. AlgaeDICE does not perform well for offline pretraining.
>
> We believe we have addressed the AWR point in the common response: AWAC empirically performs much better for both offline learning and online fine-tuning than AWR. Additionally, the novelty of this paper is in the analysis of the online fine-tuning problem and an algorithm catered to this specific setting. Are there any other concerns we can help address during the discussion phase?

---

> ### Author Response · Authors · 2020-11-24
> **Final questions?**
>
> There is a little more time left in the rebuttal period, so we have one more chance for discussion. We would appreciate any feedback or questions on the new experiments and our response. Are there any final questions we can help answer?

---

### Official Review · AnonReviewer2 · 2020-10-28
**An incremental improvement on combining offline data with online learning**

**Rating:** 6
**Confidence:** 4

**Review:**

The paper describes an approach to use offline data to accelerate the
online learning process of reinforcement learning.
The proposed approach, called AWAC, uses dynamic programming to train
a critic and supervised learning to trained a constrained actor.
The idea is to use a static dataset of experience tuples for
pre-training and use some online interactions to learn the optimal
policy for the current task.
For the policy improvement step they optimize the policy to maximize
the estimated Q-value function but constrained to remain close to the
actions observed in the data, using the Kulback-Leibler divergence.
This constrain is incorporated in the optimization and solve using the
Lagrangian.
The proposed approach is very similar to AWR, however, instead of
estimating the value function of the behavior policy with a Monte
Carlo approach, they estimate the Q function of the current policy via
bootstrapping.
AWAC is tested in several MuJoCo simulator problems, three shown in
the main paper (in-hand rotation of a pen, opening a door by
unlatching the handle, and picking up a sphere and relocating it in a
target location), and others shown in the appendix.

Positive:
- Use previously collected data for online learning is certainly
relevant for reinforcement learning is domains such as robotics. The
proposed approach shows that a slight change in how to evaluate the
Q-function to estimate the advantage function can produce a
significant difference

Negative:
- The paper follows very closely the description of AWR with only
a slight variation in how to evaluate the advantage function.
- It is not clear how close the static dataset has to be to a "good"
policy and how it affects the learning process.

---

> ### Author Response · Authors · 2020-11-14
> **Initial response to R2**
>
> Thank you for the review, and please see the common response on the comparison with AWR. Further experiments on varying the quality of the dataset are in progress and should be added to the paper shortly.

---

> ### Author Response · Authors · 2020-11-24
> **D4RL experiments with varying data quality**
>
> We have finished running the experiments we alluded to earlier, varying the quality of the initial datasets in response to the following suggestion:
>
> > It is not clear how close the static dataset has to be to a "good" policy and how it affects the learning process.
>
> We have updated the paper to add in appendix A.7 the results of running AWAC on the varied data quality (random, medium, medium-expert, and expert) datasets included in D4RL, first by training offline and then fine-tuning online. The performance of BEAR [1] is attached as reference. We attempted to fine-tune BEAR online using the same protocol as AWAC but the performance did not improve and often decreased; thus we report the offline performance. All performances are scaled 0-100, where 0 is a random policy and 100 is an expert policy, as is standard for D4RL.
>
> The results are presented in Figure 8. First, we observe that AWAC offline is competitive with BEAR offline. Then, AWAC is able to make progress in solving the tasks with online fine-tuning, even when initialized from random data or “medium” quality data. We found that AWAC is in fact able to improve online in almost every setting. When starting with lower quality data, there is more room for online improvement so the gap between BEAR is higher. But AWAC is also able to take advantage of higher quality datasets to start at a higher initial performance.
>
> Please let us know if there are any other concerns we can help address.
>
> [1] Stabilizing Off-Policy Q-Learning via Bootstrapping Error Reduction. Kumar et al. 2019.

---

> > ### Comment · AnonReviewer2 · 2020-11-25
> > **After all the comment I'll keep my score**
> >
> > I have read all the reviews and the author's responses. Even if there is the proposed method presents a slight modification with respect to AWR, the authors have made a clear case of its benefits, so I will keep my score of 6: Marginally above acceptance threshold

---

### Official Review · AnonReviewer3 · 2020-11-02
**A fairly good idea with not so high technical depth**

**Rating:** 6
**Confidence:** 3

**Review:**

Summary:

In this paper, the authors intend to accelerate on-line reinforcement learning with off-line datasets. To achieve this goal, they propose an algorithm called advantage weighted actor-critic (AWAC), which uses an implicit constraint to reduce accumulated bootstrapping error when doing off-line training and reduce the conservation when doing on-line fine-tuning. The experiments show that the proposed method can learn difficult, high-dimensional, sparse reward dexterous manipulation problems from human demonstrations and off-policy data.

Pros:
1. The organization is clear and related work is sufficient. This paper introduces the off-line training and on-line fine-tuning problem and its challenge by gradually summarize the existing studies, and with sufficient investigation of existing work, it is easy for readers to dive into the problem
2. The proposed method is easy to implement. Thanks to the implicit formulation of the constraint, the proposed AWAC algorithm can sample directly from off-line datasets without fitting a parametric model. This strategy is especially advantageous when the dimensionality is high. Besides, since this algorithm does not need the parametric model any more, it is more friendly for users who are not familiar with the off-line dataset.
3. The details of experimental settings are provided, and thus there should be no issues with the repeatability of the experiments.

Cons:
1. It is doubtful for the contribution on online fine-tuning. The constraint is designed for off-line training because off-line training has the problem of bootstrapping error accumulation. There is no judgment or proof that online training also suffers from the bootstrapping problem. How can the authors claim their contribution on on-line learning with the constraint?
2. The technical depth is not high. It seems the only contribution of this paper is to combine the constraint with an unparameterized strategy. But I would like to say that this is not a big problem, as many influential reinforcement learning algorithms are straightforward but have very significant improvement, e.g., DDQN and PPO.
3. The writing needs polish. E.g., in Equation (4), it should be 'max' instead of 'argmax', and there exists non-English spelling in Section 3.1.

---

> ### Author Response · Authors · 2020-11-14
> **Initial response to R3**
>
> We thank the reviewer for their review.
>
> > How can the authors claim their contribution on on-line learning with the constraint?
>
> The issue is that for effective online fine-tuning, we first need effective offline pre-training that can then be continued online. Empirically, ours is the only method in our experiments that demonstrates this capability. For instance, see how SAC with BC pretraining has a characteristic dip in easier problems (Figure 2 plot 2) and fails to learn in harder problems (Figure 3) because the initialization from pre-training does not persist through fine-tuning. For effective and efficient online RL from prior data, we need a method that achieves decent offline performance and preserves that performance through online exploration, so that exploration data during the online phase is also meaningful.
>
> What allows AWAC to actually fine-tune online while retaining offline pre-training performance is not just the particular form of the constraint -- other algorithms such as BRAC (Wu et al 2019) also use the KL constraint -- but the way we enforce this constraint implicitly. This allows us to actually perform the update without any explicit behavior modeling, which as we discuss in Section 3.3 is the main reason for prior offline RL methods being too conservative for online finetuning. In summary AWAC is able to combine the best of both worlds - good offline pretraining like BRAC (Wu et al), BEAR (Kumar et al) which is much better than standard off-policy RL algorithms like SAC (Haarnoja et al), and good online finetuning like SAC, performing much better than the offline RL algorithms like BRAC (Wu et al), BEAR (Kumar et al), which are too conservative due to imperfect behavior modeling.
>
> > Writing suggestions
>
> Thank you for these suggestions. We changed the notation in equation (4). Additionally, we have updated the paper with corrections in section 3, but please let us know if we have missed something.

---

### Official Review · AnonReviewer5 · 2020-11-03
**interesting paper but some concerns**

**Rating:** 4
**Confidence:** 5

**Review:**

This paper studies shorting coming of existing off-policy methods when it comes to prior data and fine-tuning and shows that those existing methods can't effectively utilize previously collected data with online updates. To address this problem, they propose to constraint policy updates with respect to behavioral policy. Their proposed method is built mainly on the top of AWR [1].

- There is a significant similarity between the proposed method and AWR [1]. The main difference that I can see is AWR doesn't have a fine-tuning step but this paper does. Can authors list their differences with AWR?

- CRR [3] is very similar to the proposed method in this paper as well, what are the differences? ( I disagree with the authors that CRR is a concurrent work with this paper)

-  This paper claims that AWAC can utilize various types of prior data without any changes to their method and it is agnostic to the type of behavioral policy (e.g. random vs. expert). I don't see how this is the case. For example, if data were collected by a random policy, this method constrains the current policy to be a uniform one.

- Experiments don't make a case for this method, i.e. AWAC is only better than others in relocate-binary-v0 and barely better in Walker2d-v2.  ABM almost gets the same performance.

- Fine-Tuning with random policy data is not convincing either. I'd suggest running some experiments with D4RL [5] dataset like *-medium-expert, *-medium-replay, and *-random to build a better case for your proposed method.

- One reason that existing RL methods can't fully utilize previously collected data with online updates is the distribution shift between the policy (a.k.a behavioral policy) that collected the data and the learned policy [2, 3]. Can the authors comment on this and discuss how the proposed method addresses the distribution shift problem that likely the root cause of this problem? A section in the paper about this will help to improve this paper.

- Related work section needs significant improvement. It only lists various papers without any useful discussion.

Even though I found this paper interesting, I'm still not convinced about the contribution of this paper. Given similarity and overlap with previous works [1,3], more experiments could be helpful to see how this method is better than others and why one should select this method vs. others.

Minor comments:

It seems there are various problems with your bibtex, for example, "Behavior Regularized Offline Reinforcement Learning" wasn't published in iclr and author names are mixed with other information like in  "A Generalized Path Integral Control Approach to Reinforcement Learning" "usc" in author names and "Web Services" appeared with author names "P3O: Policy-on Policy-off Policy Optimization."

[1] Xue Bin Peng, Aviral Kumar, Grace Zhang, and Sergey Levine. Advantage-Weighted Regression: Simple and Scalable Off-Policy Reinforcement Learning. sep 2019.

[2] Sergey Levine, Aviral Kumar, George Tucker, and Justin Fu. Offline Reinforcement Learning: Tutorial, Review, and Perspectives on Open Problems. Technical report, 2020

[3 Rasool Fakoor, Pratik Chaudhari, and Alexander J Smola. P3O: Policy-on Policy-off Policy Optimization. In Conference on Uncertainty in Artificial Intelligence (UAI), 2019.

[4] Ziyu Wang, Alexander Novikov, Konrad Zołna, Jost Tobias Springenberg, Scott Reed, Bobak Shahriari, Noah Siegel, Josh Merel, Caglar Gulcehre, Nicolas Heess, and Nando De Freitas. Critic Regularized Regression. 2020.

[5] Justin Fu, Aviral Kumar, Ofir Nachum, George Tucker, Sergey Levine. D4RL: Datasets for Deep Data-Driven Reinforcement Learning, 2020

---

> ### Author Response · Authors · 2020-11-14
> **Initial response to R5**
>
> We thank the reviewer for the thorough review. Specific concerns are addressed below.
>
> > differences with AWR
>
> The key difference between AWAC and AWR is the value estimation: in AWAC, we use a fully off-policy estimate of A^pi, while AWR estimates A^beta through generalized advantage estimation. While this may seem like a somewhat minor modification, do observe in the experiments that AWAC is significantly more sample efficient than AWR in all tasks. Additionally, we study the fine-tuning scenario which AWAC is especially designed for.
>
> > Critic regularized regression (CRR)
>
> The CRR paper itself states that our work is concurrent: “Concurrently to our work, [25] proposed Advantage Weighted Actor Critic (AWAC) for accelerating online RL with offline datasets. Their formulation is equivalent to CRR with exponential filtering.”  - https://arxiv.org/abs/2006.15134. The eventual algorithm is very similar, but the derivations are different and we additionally study the fine-tuning scenario beyond offline RL, and provide a systematic set of controlled experiments to understand the challenges.
>
> > Experiments don't make a case for this method
>
> We believe our results are actually very strong. As you point out, our method is the only one that can solve relocate-binary-v0, which is a very difficult task involving 30 action dimensions and sparse rewards. It is additionally the single best performing method for door-binary-v0 with a final success rate of 95%, with the next best performing method ABM hovering around 50% success rate. In pen-binary-v0, ABM is slightly worse and all other methods are much worse. And while other methods perform comparably to AWAC at some tasks of the easier MuJoCo benchmark tasks, none consistently perform best like AWAC does.
>
> > Can AWAC handle arbitrary off-policy data?
>
> The parameter beta allows deviation from the behavior policy, so AWAC may improve significantly beyond the behavior policy. In addition, we have an experiment (Figure 4) where AWAC is shown to learn a policy with the behavior policy being a uniform random policy.
>
> > experiments with D4RL [5] dataset
>
> Thank you for this suggestion, we are now conducting additional experiments with varying data quality in D4RL and we will update the paper with those shortly.
>
> > Bibtex issues
>
> Thank you for pointing out these issues, they have now been resolved and the paper updated.

---

> ### Author Response · Authors · 2020-11-24
> **D4RL experiments with varying data quality**
>
> We have finished running the remaining experiments requested following your suggestion:
>
> > I'd suggest running some experiments with D4RL [5] dataset like *-medium-expert, *-medium-replay, and *-random to build a better case for your proposed method.
>
> The experiment is now included in appendix A.7: the results of running AWAC on the varied data quality (random, medium, medium-expert, and expert) datasets included in D4RL, first by training offline and then fine-tuning online. The performance of BEAR [1] is attached as reference. We attempted to fine-tune BEAR online using the same protocol as AWAC but the performance did not improve and often decreased; thus we report the offline performance. All performances are scaled 0-100, where 0 is a random policy and 100 is an expert policy, as is standard for D4RL.
>
> The results are presented in Figure 8. First, we observe that AWAC is competitive with BEAR offline. Then, AWAC is able to make progress in solving the tasks with online fine-tuning, even when initialized from lower quality "random" or “medium” quality data.
>
> Please let us know if there are any other concerns that we can help address, with this experiment or the issues raised previously.
>
> [1] Stabilizing Off-Policy Q-Learning via Bootstrapping Error Reduction. Kumar et al. 2019.

---

> ### Author Response · Authors · 2020-11-24
> **Final questions?**
>
> There is a little more time left in the rebuttal period, so we have one more chance for discussion. We would appreciate any feedback or questions on the new experiments and our response. Are there any final questions we can help answer?

---

### Author Response · Authors · 2020-11-14
**Common response**

We thank all the reviewers for their constructive comments. We will reply to the common issue raised about novelty beyond prior work in this thread and then reviewer-specific comments in their own thread.

> Novelty beyond prior work

It is indeed true that prior works have proposed similar algorithms. But we highlight two important contributions of our paper that are unique. First, we study the setting of online fine-tuning after offline pre-training in detail, identifying three key issues that plague prior algorithms in this setting, preventing these algorithms from actually solving the problem. Second, we empirically show that designing an algorithm without these issues results in an algorithm that can in fact take advantage of prior data and then quickly fine-tune with a small amount of online interaction. In comparison, the closest algorithm to AWAC, advantage weighted regression (AWR) mentioned by reviewers 1, 2 and 5, performs very poorly in this setting due to the difference in value function learning. AWR estimates the Q-value of the behavior policy, using generalized advantage estimation, while AWAC estimates Q^pi of the current policy using fully off-policy bootstrapping. This difference allows AWAC to learn a better policy in the offline phase, and then be more sample efficient in the online phase. As seen from our empirical results, this makes a huge difference in algorithm performance. For instance, in Figure 3, AWR obtains a final performance of 53%, 0% and 0% on the dexterous manipulation tasks of pen, door and relocate respectively while AWAC obtains a final performance of 97%, 95%, and 55%. The value estimation results in the difference between making no progress on the two harder dexterous manipulation tasks and being able to solve them convincingly.

---

### Decision · Program_Chairs · 2021-01-07
**Final Decision**

**Decision:**

Reject

**Comment:**

This paper proposed a new method improving online reinforcement learning using offline datasets. Three reviewers suggested (borderline) acceptance and two did rejection. The main concerns of reviewers are (a) limited/incremental novelty (from all reviewers) and (b) limited experiments (from three reviewers). AC also agrees that the authors' response for novelty beyond the prior works, e.g., AWR (and CRR), is not convincing enough (although their goals/settings are different). AC also thinks that more discussion, analysis and results when offline datasets are poor (e.g., far from experts) are necessary to meet the high standard of ICLR (the authors provided some, but AC thinks it is not convincing enough). Hence, AC recommend rejection.